# The Role of Tumor Biomarkers in Tailoring the Approach to Advanced Ovarian Cancer

**DOI:** 10.3390/ijms252011239

**Published:** 2024-10-19

**Authors:** Noemi Tonti, Tullio Golia D’Augè, Ilaria Cuccu, Emanuele De Angelis, Ottavia D’Oria, Giorgia Perniola, Antonio Simone Laganà, Andrea Etrusco, Federico Ferrari, Stefania Saponara, Violante Di Donato, Giorgio Bogani, Andrea Giannini

**Affiliations:** 1Department of Maternal and Child Health and Urological Sciences, Policlinico Umberto I, Sapienza University of Rome, 00161 Rome, Italy; noemi.tonti@uniroma1.it (N.T.); ilaria.cuccu@uniroma1.it (I.C.); emanuele.deangelis@uniroma1.it (E.D.A.); giorgia.perniola@uniroma1.it (G.P.); violante.didonato@uniroma1.it (V.D.D.); 2Obstetrics and Gynecological Unit, Department of Woman’s and Child’s Health, San Camillo-Forlanini Hospital, 00152 Rome, Italy; ottaviadr@gmail.com; 3Unit of Obstetrics and Gynecology, “Paolo Giaccone” Hospital, Department of Health Promotion, Mother and Child Care, Internal Medicine and Medical Specialties (PROMISE), University of Palermo, 90127 Palermo, Italy; antoniosimone.lagana@unipa.it (A.S.L.); etruscoandrea@gmail.com (A.E.); 4Department of Clinical and Experimental Sciences, University of Brescia, 25123 Brescia, Italy; 5Division of Gynecology and Obstetrics, Department of Surgical Sciences, University of Cagliari, 09124 Cagliari, Italy; saponara.stef@gmail.com; 6Gynecological Oncology Unit, Fondazione IRCCS Istituto Nazionale dei Tumori, 20133 Milan, Italy; bogani.giorgio@gmail.com; 7Unit of Gynecology, Department of Surgical and Medical Sciences and Translational Medicine, Sant’Andrea Hospital, Sapienza University of Rome, 00189 Rome, Italy; andrea.giannini@uniroma1.it

**Keywords:** ovarian cancer, gynecologic oncology, *BRCA*, HRD, molecular classification, target therapies, Parpi

## Abstract

Growing evidence has demonstrated the role of mutations of tumor biomarkers in diagnosing and treating epithelial ovarian cancer. This review aims to analyze recent literature on the correlation between tumor biomarkers and chemotherapy in nonmucinous ovarian cancer, providing suggestions for personalized treatment approaches. An extensive literature search was conducted to identify relevant studies and trials. *BRCA1/2* mutations are central in homologous recombination repair deficiency (HRD) in ovarian cancer, but several other genetic mutations also contribute to varying cancer risks. While the role of MMR testing in ovarian cancer is debated, it is more commonly linked to non-serous ovarian cancer, often associated with Lynch syndrome. A significant proportion of ovarian cancer patients have HRD, affecting treatment decisions in both first-line (especially in advanced stages) and second-line therapy due to HRD’s connection with platinum-based therapy and PARP inhibitors’ response. However, validated genetic tests to identify HRD have not yet been universally implemented. There is no definitive therapeutic algorithm for advanced ovarian cancer, despite ongoing efforts and multiple proposed tools. Future research should focus on expanding the utility of biomarkers, reducing resistance, and increasing the actionable biomarker pool.

## 1. Introduction

Ovarian cancer (OC) is the primary cause of death among all gynecological malignancies in developed countries. In 2020, the global incidence of OC was documented at 313,959 new cases, with a consequential 207,252 recorded deaths [1]. The European projected incidence and mortality rates were 15.5 and 10.3 per 100,000, respectively [2]. The disease stage at diagnosis is the leading prognostic factor and the 5-year overall survival is globally around 46% [3]. Most patients receive the diagnosis at an advanced stage, which is frequently associated with increased mortality. The most frequent histologic type of OC originates from the epithelium and the established approach for treating this cancer at the advanced stage (International Federation of Gynecology and Obstetrics [FIGO] stage II–IV) involves optimal debulking surgery followed by platinum-based chemotherapy. Despite the initial responsiveness to chemotherapy, a significant number of patients experience relapse within 3 years [4,5,6,7,8]. Fortunately, the landscape of OC treatment has undergone a significant transformation with the advent of targeted therapies. In particular, poly (ADP-ribose) polymerase inhibitors (PARPis) have emerged as crucial maintenance therapies for OC in both initial and relapsed disease scenarios. Indeed, approximately 20% of epithelial ovarian cancer (EOC) patients exhibit mutations in breast cancer susceptibility gene 1 (*BRCA1*) and gene 2 (*BRCA2*), contributing to homologous recombination deficiency (HRD).

Furthermore, up to 50% of high-grade serous OC manifests HRD [9]. These patients demonstrate genomic instability characterized by one or more defects in DNA repair pathways. In this setting, PARPis play a pivotal role by impeding the repair of DNA single-strand breaks, inducing cell death through synthetic lethality in patients with *BRCA* mutations or HRD [10]. While more treatment options are becoming available for OC, determining the best therapy for patients can still be challenging for clinicians. For this reason, the essentiality of biomarker testing resides in discerning patients who are predisposed to benefit from maintenance therapy with PARPi. This scientific review aims to explore the role of tumor biomarkers in the decision-making process for therapeutic approaches in advanced ovarian cancer (OC) to ensure that patients receive the best-tailored therapy and the current target therapies for adjuvant treatment.

## 2. Exploring DNA Repair Pathways: The Impact of *BRCA1/2* and Other Mutations on Cancer Susceptibility and Treatment

### 2.1. The Fundamentals of Homologous Recombination Repair

The integrity of our genetic material is crucial to the survival and proper functioning of all living organisms. DNA damage is frequent during the cell life cycle and can result in a single-strand break (SSB) or double-strand break (DSB). Understanding the mechanisms by which these breaks are repaired is a fundamental aspect of molecular biology and genetics. Humans employ at least five major DNA repair pathways, each active at various stages of the cell cycle: base excision repair (BER), nucleotide excision repair (NER), mismatch repair (MMR), homologous recombination and non-homologous end joining (NHEJ). Base excision repair specifically addresses single-strand breaks, while homologous recombination and non-homologous end joining represent the primary pathways for mending double-strand breaks. The malfunction, reduction or absence of proteins implicated in these repair pathways is linked to mutagenesis, toxicity, cancer disease and cell death [11,12]. Homologous recombination is one of the most intricate and high-fidelity mechanisms for repairing DSBs. It includes several proteins such as *BRCA1* and *BRCA2*, proteins of the *Mre11-Rad50- Nbs1 (MRN) complex, CtIP, MRE11, RAD51, ATM, H2AX, PALB2, RPA, RAD52* and the Fanconi anemia pathway proteins [13,14]. When homologous recombination repair (HRR) is dysfunctional in cells due to *BRCA1* or *BRCA2* deficiency, alternative pathways such as non-homologous end joining are activated. However, these alternative pathways may lead to inaccurate repairs, causing the accumulation of extra DNA amplifications or deletions. This can result in chromosomal instability, elevating the susceptibility to developing cancer [15].

### 2.2. The Pivotal Role of BRCA1 and BRCA2 Mutations

Genetic factors substantially predispose individuals to OC, with *BRCA1* and *BRCA2* mutations being well-established risk factors. A female’s lifetime risk of developing OC is 1.3% [16]. Approximately 20% of cases are linked to germline mutations, with most ascribed to *BRCA1* or *BRCA2* mutations [9,17]. These genes codify leading proteins involved in the homologous recombination of DSB [18]. Their mutations compromise the cellular capacity to repair DNA double-strand breaks, resulting in genomic instability and facilitating tumorigenesis. Identifying *BRCA1* and *BRCA2* mutations holds significant clinical importance for individuals diagnosed with OC. In fact, in patients with these pathogenetic mutations, OC exhibits distinct biological behavior, manifesting with different patterns of disease, an earlier average age of onset, heterogeneous responses to chemotherapy and variable prognoses [19]. Indeed, individuals with *BRCA1* and *BRCA2* mutations show improved 5-year survival, although the benefit at 10 years remains uncertain [20,21]. This might be clarified by the enhanced sensitivity to platinum-based chemotherapy and the extended disease-free periods observed in these individuals [22]. Moreover, multiple studies have established that patients with *BRCA1* and *BRCA2* mutations respond best to PARP inhibitors [23,24,25,26]. The sensitivity to PARP inhibitors is explained through synthetic lethality [10,27].

### 2.3. Beyond BRCA Mutations: Significance of RAD51, PALB2, and RAD51C Mutations

While *BRCA* mutations have long been recognized as critical genetic risk factors for breast and ovarian cancers, recent research has shed light on the importance of other genes, such as *RAD51*, *PALB2*, and *RAD51C*, in OC development and predisposition. These proteins are components of homologous recombination pathways that relate and cooperate with *BRCA1* and *BRCA2* proteins in the DNA repair process to maintain genomic stability. These homologous recombination genes together constitute a further 2% of OC cases. Tumors carrying these mutations exhibit a *BRCA*-like profile, characterized by high-grade serous histology, an increased responsiveness to platinum and improved disease-free (DFS) and overall survival (OS) rates [28,29]. A recent case–control study analyzed three *RAD51* genes in germline DNA in OC patients (*RAD51B*, *RAD51C* and *RAD51D*). They included 3429 women with invasive epithelial OC and 2772 healthy controls. The findings revealed that 0.81% of OC cases exhibited a pathogenic variant in one of these three genes compared with 0.11% in controls [30]. Moreover, Hodgson et al. (2018) conducted an exploratory biomarker analysis on tumor samples from Study 19 (Olaparib maintenance therapy phase II randomized clinical trial D0810C00019; NCT00753545, 15 September 2009) to identify patients who may benefit from Olaparib beyond having *BRCA* mutations. In this study, 95/209 patients (45.5%) were classified as having a *BRCA* wild-type tumor, while 21/95 patients had *BRCA* wild-type tumors with loss-of-function HRR mutations; among these, 12 received Olaparib and 9 received placebo. Of the remaining 74 patients, 16 were classified as HRR unknown, 58 as no detectable HRR mutations, 25 were randomized to Olaparib and 33 to placebo. This exploratory analysis showed that Olaparib provides a greater progression-free survival (PFS) advantage in HRR-mutated patients without a *BRCA* mutation (hazard ratio (HR) 0.21, 0.04–0.86) compared to patients without detectable *BRCA* or HRR mutations (HR 0.71, 0.37–1.35). Patients with tumors and loss-of-function mutations in HRR genes, distinct from *BRCA1/2*, might represent a small population who could derive benefit from Olaparib [31]. Even if these genes seem to play a crucial role in the susceptibility of OC, current European Society for Medical Oncology (ESMO) guidelines assert that there is insufficient evidence to establish the clinical validity of individual or panels of non-*BRCA* HRR mutations for predicting the response to PARPis [32].

### 2.4. Lynch Syndrome: Role of Mutations in Mismatch Repair Genes and MMR Testing

Lynch syndrome (LS) is the second most common cause of hereditary OC and is characterized by an increased susceptibility to multiple cancer types, primarily colorectal, endometrial and ovarian cancer. It is caused by a mutation in the MMR genes (*MLH1*, *MSH2*, *MSH6*, *PMS2*), which constitute a system that plays a crucial role in maintaining genomic stability [33]. In Lynch patients, the lifetime risk of OC ranges from 8 to 20 percent (depending on the MMR gene involved). On the other hand, about 2–3% of cases of OC are related to Lynch syndrome [34,35]. Additionally, while high-grade serous carcinoma is the primary histological type of OC linked to *BRCA* mutations, Lynch syndrome is more frequently observed in non-serous ovarian cancers, such as endometriosis-derived clear-cell and endometrioid ovarian carcinomas [36]. The role of testing MMR in OC is controversial. Indeed, while patients with colorectal cancer received a universal screening for Lynch syndrome, current guidelines do not recommend the same approach for OC. Indeed, due to the low rate of OC related to LS, universal screening may not be cost-efficient. Nevertheless, MMR testing in OC seems to have both prognostic value and therapeutic implications. MMR testing can be performed using several methodologies, including immunohistochemistry (IHC) for MMR protein expression and PCR-based MSI testing. Typically, Lynch patients with OC have an earlier age of onset, with a mean age at diagnosis of 48 years and generally, OC presents at an early stage at diagnosis (FIGO stage I–II) [37,38]. Moreover, data suggest that even in advanced-stage OC, survival may be better in mismatch repair deficient (dMMR) carriers compared to *BRCA*-mutated carriers or the general population. Indeed, in a study conducted by Niskakoski et al., it has been shown that none of the LS-associated OC had mutations in *TP53*, which is generally associated with a worse prognosis [39]. For that reason, testing the MMR status in OC has been demonstrated to have a significant prognostic value. On the other hand, even if the prognosis in these patients has been demonstrated to be better, data about the predictive sensitivity to chemotherapy or PARPi are missing. However, in recent years, there have been significant advancements in the treatment of Lynch syndrome-associated cancers, as well as other microsatellite instability-high (MSI-H) and MMRd cancers. Indeed, a study published in 2015 by Le et al. demonstrated that a monoclonal antibody anti-PD-1 (Pembrolizumab) improved objective response rates and overall disease control rates in patients with advanced MMRd/MSI-H cancers [40]. Due to this result, in 2017, the US Food and Drug Administration approved Pembrolizumab for tumors with MSI-H or dMMR, including ovarian cancer, highlighting the clinical utility of MMR testing in guiding immunotherapy. Finally, due to the hereditary nature of some MMR defects, MMR testing also has implications for genetic counseling and screening. Detecting Lynch syndrome in patients with OC allows for the implementation of surveillance strategies for at-risk family members, potentially reducing cancer incidence and mortality through early detection and intervention. For instance, MMR testing is a valuable tool in the management of OC, providing diagnostic, prognostic and therapeutic insights, and it could be considered in non-serous OC, which is more frequently related to LS (Table 1).

### 2.5. Role of Biomarker Testing in Optimal Therapeutic Decisions

Tumor biomarker testing is essential for guiding decisions regarding the maintenance therapy of advanced OC. As previously discussed, clinical trials revealed that maintenance therapy with PARP inhibitors demonstrated the most significant advantages in newly diagnosed OC patients with either *BRCA* mutations or positive results for HRD [23,24,25,26]. Current guidelines suggest conducting germline testing for all EOC patients at the time of diagnosis. Additionally, tumor testing for somatic *BRCA* mutations is recommended for individuals without identified germline *BRCA* mutations. Moreover, germinal *BRCA* testing is essential to predict the risk of developing other related cancers, and it permits a genetic risk evaluation of first- or second-degree blood relatives [41,42]. The pivotal role of the germline *BRCA* test is also connected to the possibility of identifying people who may derive advantages from genetic counseling and the implementation of risk-reducing strategies. Indeed, genetic counseling should be accessible to individuals undergoing genetic testing during their primary diagnosis, particularly those with a family history of breast or ovarian cancer [42,43]. Interpreting germline test results for *BRCA1* and *BRCA2* typically involves assessing whether specific mutations or variants are present. Indeed, the pathogenicity of each variant is classified by the laboratory into one of five categories (pathogenic, likely pathogenic, variant of uncertain significance [VUS], likely benign, benign), using information available at the time [44]. The classification for many variants continues to be updated, especially for VUS as more evidence from research becomes available [45]. Due to the high rate of VUS, patients must be informed that such findings are expected and that many VUS will be reclassified over time. The VUS detection should be interpreted as a result lacking conclusive information and should not impact the clinical management of the patient. Decisions regarding risk management should be primarily guided by personal and/or family history. Additionally, the presence of a VUS can pose challenges for relatives, as it does not provide clear guidance for implementing surveillance and prevention strategies within the family [46,47]. In addition to *BRCA* germline/somatic testing, HRD testing is fundamental in the newly diagnosed OC setting to guide clinical decisions for maintenance therapy. As it has been widely discussed, HRD is defined by the existence of *BRCA* mutations or genomic instability, determined by the occurrence of loss of heterozygosity (LOH), telomeric allelic imbalance, and large-scale state transitions in the genome [43]. Validated commercially available assays for test HRD status are Myriad MyChoice^®^ CDx (Myriad Genetics, Inc., Salt Lake City, UT, USA) and FoundationOne CDxTM (Foundation Medicine, Inc., Cambridge, MA, USA). PRIMA and PAOLA-1 trials [23,24] in the newly diagnosed OC setting employed MyChoice^®^ CDx, a diagnostic tool that assesses the existence of a *BRCA*m and/or genomic instability (LOH, telomeric allelic imbalance and large-scale state transitions). In the myChoice assay, HRR deficiency is typically identified either by the presence of *BRCA1/2m* or when a tumor exceeds a specified threshold of genomic instability score (GIS). A GIS threshold of 42 was the score in PAOLA-1/ENGOT-ov25 and PRIMA/ENGOTov26 (21,22). The ATHENA trial [25] used FoundationOne CDxTM, which tests for the presence of a *BRCA*m and LOH. However, there is variability among HRR deficiency tests in terms of the specific genomic features they assess and the methods employed to establish the thresholds defining HRR deficiency. Consequently, these assays are not interchangeable [43]. As can be seen in these trials, HRD-induced genomic scars have been assessed by centralized next-generation sequencing (NGS)-based assays (Myriad MyChoice^®^ CDx and FoundationOne CDxTM). This implies that HRD testing needs to be outsourced, potentially exerting a substantial influence on the effectiveness of molecular tests in clinical applications. In a retrospective study conducted by Pepe et al., they found a possible solution by using in-house testing for HRD that confirmed a high agreement compared to the external gold standard [48]. Finally, in a study conducted by Leary et al., the innovative role of circulating tumor DNA (cDNA) obtained from ascites was considered for detecting the HRD status using the Single Nucleotide Polymorphism Array, estimating somatic copy number alterations (SCNA). They concluded that SCNA analysis on ascitic cDNA is viable and can detect the same HRD scar as tumor testing [49].

## 3. Clinical Outcomes in Homologous Recombination Deficiency: Key Insights from Clinical Trials in First-Line and Second-Line Therapies

### 3.1. Overview

PARPis disrupt the repair process of DNA SSBs, and in OC, they are linked to *BRCA* mutations or HRD cell death. They induce death by disrupting the efficiency of DNA repair mechanisms [10]. The advent of these new therapies has completely changed the management of OC in both first-line and recurrent disease scenarios. Over the last decade, PARP inhibitors have undergone extensive research, showing highly promising results. The FDA has approved three PARP inhibitors—Olaparib, Rucaparib, and Niraparib—for multiple indications in OC.

### 3.2. Olaparib in First-Line Maintenance Therapy

Based on the findings from the Phase III SOLO1 trial, Olaparib became the first PARP inhibitor to receive approval for first-line maintenance monotherapy in the United States [26]. In this study, 391 participants were randomly allocated to two groups: Olaparib (*n* = 260) at a dosage of 300 mg twice daily and placebo (*n* = 131). Individuals in both cohorts had been diagnosed with FIGO stage III–IV, high-grade, *BRCA*-associated serous or endometrioid ovarian, fallopian tube or primary peritoneal cancers. All participants demonstrated partial or complete clinical response to platinum-based chemotherapy. Among them, 388 patients had centrally confirmed germline *BRCA* mutations, and none had been administered Bevacizumab. Following a median follow-up of 41 months, Olaparib exhibited a significantly lower 3-year rate of disease progression or death compared to the counterpart (60% vs. 27%, HR: 0.30, 95% CI: 0.23–0.41, *p* < 0.0001), with no adverse impact on health-related quality of life [26].

### 3.3. Long-Term Efficacy of Olaparib

A post hoc analysis from the SOLO-1 after 5 years of follow-up showed that the median PFS was 56 months (95% CI: 41.9 months not reached) in the Olaparib group versus 14 months in the placebo group (HR: 0.33; 95% CI: 0.25–0.43) [50]. Finally, after a 7-year follow-up, a descriptive analysis revealed a notable enhancement in OS with Olaparib compared to placebo (median OS not reached vs. 75.2 months; HR 0.55, 95% CI 0.40–0.76; *p* = 0.0004) [51].

### 3.4. Maintenance Therapy Regardless Biomarkers Status: Niraparib in First Line

Subsequently, Niraparib has received FDA approval for first-line monotherapy, regardless of the tumor’s *BRCA* status, based on the results of the Phase III PRIMA trial [23]. Indeed, differently from SOLO-1, PRIMA included also patients without deleterious mutations in *BRCA1/2*. The results demonstrated a significant enhancement in PFS with Niraparib monotherapy for the overall population, regardless of the presence or absence of HRD. In this clinical trial, 733 patients diagnosed with advanced OC were randomly allocated in a 2:1 ratio to receive daily doses of either Niraparib or a placebo following a positive response to platinum-based chemotherapy. The results revealed a statistically significant improvement in PFS among patients treated with Niraparib, both in the HRD and overall populations. In the HRD subgroup, the median PFS was 21.9 months for Niraparib recipients, in contrast to 10.4 months for the placebo group (HR: 0.43; 95% CI: 0.31–0.59; *p* < 0.0001). Meanwhile, in the overall population, the median PFS was 13.8 months for those receiving Niraparib and 8.2 months for placebo (HR: 0.62; 95% CI: 0.50–0.76; *p* < 0.0001). At the 24-month interim analysis, the OS rate was 84% in the Niraparib group and 77% in the placebo group (HR: 0.70; 95% CI: 0.44–1.11) [23]. Therefore, the findings from the PRIMA trial demonstrated the positive impact of Niraparib treatment on PFS in individuals diagnosed with advanced OC, regardless of their HRD status.

### 3.5. Maintenance Therapy Regardless Biomarkers Status: Recuparib in First Line

In the ATHENA-MONO trial, 538 patients were randomized to receive maintenance Rucaparib or placebo for up to 2 years or until disease progression, death or the occurrence of unacceptable toxicity [25]. In this study, patients with newly diagnosed stage III–IV high-grade OC, who underwent surgical cytoreduction and showed a response to first-line platinum-based chemotherapy, were included, regardless of their biomarker status. Stratification based on HRD status, determined by the FoundationOne CDxTM next-generation sequencing assay, was implemented. Their primary objective was to assess PFS in patients with HRD-positive tumors (defined as having a BRCA mutation and/or a high genomic loss of heterozygosity [LOH] score [≥16%]) and in the overall population. Following a median follow-up of approximately 26 months, the Rucaparib group demonstrated a significantly longer median PFS compared to the placebo group, both in patients with HRD-positive tumors (28.7 vs. 11.3 months; HR0.47; 95% CI 0.31–0.72; *p* = 0.0004) and in the overall population (20.2 vs. 9.2 months; HR 0.52; 95% CI 0.40–0.68; *p* < 0.0001). Although benefits were also observed in the HRD-negative group (12 vs. 9.1 months, HR 0.65, 95% CI 0.450.95), they were comparatively less than those in the HRD-positive group [25]. Despite these interesting results, Rucaparib lacks regulatory approval in the front-line setting.

### 3.6. Combined Therapy: Olaparib and Bevacizumab

Finally, another significant clinical trial has evaluated the combination of two agents with synergistic action, PARPi and the antiangiogenic agent Bevacizumab. Indeed, the results of the PAOLA-1 supported the FDA’s approval of Olaparib in combination with Bevacizumab for maintenance therapy in individuals with newly diagnosed, advanced, high-grade OC who have shown a positive response to first-line platinum-based chemotherapy plus Bevacizumab [24]. In this trial, patients were randomly assigned to two maintenance groups: Bevacizumab alone (15 mg/kg every three weeks) or Bevacizumab with Olaparib (300 mg twice daily). The trial demonstrated a significant PFS advantage with the addition of maintenance Olaparib, particularly in patients with advanced OC who had undergone first-line standard therapy, including Bevacizumab. This improvement was observed even in patients with HRD-positive tumors, irrespective of *BRCA* mutations. After a median follow-up of 22.9 months, a statistically significant increase was noted in the median PFS for patients receiving Olaparib compared to the placebo (22.1 months vs. 16.6 months; HR: 0.59; 95% CI: 0.49–0.72; *p* < 0.001). The HR (Olaparib Group vs. Placebo Group) for disease progression or death was 0.33 (95% CI: 0.25–0.45) in patients with HRD-positive tumors carrying *BRCA* mutations (median PFS: 37.2 months vs. 17.7 months) and 0.43 (95% CI: 0.28–0.66) in patients with HRD-positive tumors without *BRCA* mutations (median PFS: 28.1 months vs. 16.6 months). Within *BRCA*-positive OC subgroups, the combination of Olaparib and Bevacizumab improved PFS in contrast to Bevacizumab alone (37 months vs. 22 months; HR: 0.31, 95% CI: 0.20–0.47). On the other hand, in individuals with negative or unknown HRD status, the inclusion of Olaparib with Bevacizumab did not confer a benefit (median PFS was 16.9 versus 16 months with and without Olaparib, respectively; HR 0.92, 95% CI 0.72–1.17) (22). Finally, Ray-Coquard et al. published the final overall survival results from the PAOLA-1/ENGOT-ov25 trial [24], confirming the previous results. With a median follow-up of 61.7 months in the Olaparib arm and 61.9 months in the placebo arm, the median OS was 56.5 months versus 51.6 months in the intention-to-treat population (HR 0.92, 95% CI 0.76–1.12; *p* = 0.4118). Furthermore, in the HRD-positive subgroup, the addition of Olaparib to Bevacizumab was associated with prolonged OS (HR 0.62, 95% CI 0.45–0.85), resulting in a 5-year OS rate of 65.5% compared to 48.4%. The updated 5-year PFS also indicated a higher percentage of patients treated with Olaparib plus Bevacizumab without recurrence (HR 0.41, 95% CI 0.32–0.54; 5-year PFS rate, 46.1% versus 19.2%) [52] (Table 2).

### 3.7. Recurrent Disease: PARPi as a Possible Arm in Second-Line Therapy

After the diagnosis of OC, despite the initial responsiveness to chemotherapy, a significant number of patients experience relapse within 3 years (range 4–7). The possibility to use PARPi in patients with recurrent disease represents a promising therapeutic option considering the poor prognosis of these patients. Indeed, several clinical trials have demonstrated the efficacy of PARP inhibitors as a second-line therapy in OC. In Study 19, a randomized double-blind placebo-controlled phase 2 trial, 265 patients with relapsed OC who had received two or more platinum-based CT with a complete or partial response according to RECIST criteria were randomized to receive Olaparib versus placebo until disease progression. After a median follow-up of 5.6 months, the PFS in the overall population was significantly longer in the Olaparib group than in the placebo (median PFS 8.4 vs. 4.8 months; HR 0.35; 95% CI 0.25–0.49; *p* < 0.001) [53]. Moreover, a retrospective, preplanned, subgroup analysis revealed a PFS benefit in the Olaparib group in both patients with *BRCA*m (HR 0.18; 95% CI 0.10–0.31) and those without *BRCA*m (HR 0.54; 95% CI 0.34–0.85), with a greater PFS benefit observed in patients with *BRCA*m. On the other hand, at the time of the second interim analysis (58% maturity), there were no significant differences in terms of OS between the Olaparib versus placebo groups (HR 0.88 [95% CI 0.64–1.21]; *p* = 0.44) [54]. Similar results have been showed in the SOLO2/ENGOT-Ov21, a multicenter double-blind, randomized placebo-controlled phase 3 trial that evaluated the efficacy of Olaparib as a maintenance therapy in patients with platinum-sensitive relapsed OC with *BRCA1/2* mutations. The trial showed that Olaparib significantly improved PFS compared to placebo (19.1 months vs. 5.5 months, HR 0.30 [95% CI 0.22–0.41], *p* < 0.0001, respectively) [55]. In the NOVA study, 553 patients with platinum-sensitive recurrent OC with or without a germline *BRCA* mutation were randomized to receive Niraparib versus placebo. This study demonstrated that Niraparib extended PFS significantly in the entire population, regardless of the presence of *BRCA* mutations or HRD status. In particular, after a median follow-up of 16.9 months, PFS was 21.0 months vs. 5.5 months (HR 0.27; 95% CI 0.17–0.41; *p* < 0.001) in the germline *BRCA*-mutated group, whose PFS was 9.3 vs. 3.9 months (HR 0.45; 95% CI 0.34–0.61; *p* < 0.001) in patients without a *BRCA*m and finally 12.9 months vs. 3.8 months (HRD 0.38; 95% CI, 0.24–0.59; *p* < 0.001) in the non-germline *BRCA* group with HRD [56]. Additionally, in ARIEL 3, a randomized, double-blind, placebo-controlled phase 3 trial, Rucaparib demonstrated significant improvement in terms of median PFS across all patient subgroups, including *BRCA*m, HRD-positive, and the overall population. In particular, the PFS in patients with a *BRCA*m was 16.6. vs. 5.4 months (HR 0.23; 95% CI 0.16–0.34; *p* < 0.0001), in HRD-positive patients was 13.6 vs. 5.4 months (HR 0.32; 95% CI 0.24–0.42; *p* < 0.0001), and in the intention-to-treat population was 10.8 vs. 5.4 months (HR 0.36; 95% CI 0.30–0.45; *p* < 0.0001) [57]. In conclusion, the results of these studies have demonstrated that PARPi, when used as second-line therapy, significantly improves PFS, offering a valuable option for managing this challenging disease. On the other hand, it is important to note that these data are derived from patients with a recurrent disease who have not previously received PARPi as first-line therapy and who exhibit a complete or partial response to platinum-based CT according to RECIST criteria. The high rate of platinum resistance in the recurrent disease represents a major challenge for clinicians. Indeed, patients with platinum-resistant disease are generally not eligible for second-line PARPi therapy, as they are unlikely to benefit from it. Indeed, data on post-progression treatments from a retrospective study seem to suggest cross-resistance between PARPi and platinum-based chemotherapy, especially in *BRCA*m patients [58]. Furthermore, the increasing use of PARP inhibitors in the first-line setting in patients with advanced OC limits the pool of relapsed patients eligible for second-line treatment. Consequently, re-treating patients with PARP inhibitors after their initial use in the first-line setting presents unique challenges and remains a topic of debate within the scientific community. The OReO/ENGOT Ov-38, a randomized double-blind, multicenter phase 3 trial, was the first study that investigated the efficacy of Olaparib re-treatment in patients with relapsed OC who had previously benefited from PARPi maintenance therapy. The study showed that re-treatment with Olaparib significantly improved PFS compared to placebo in both the *BRCA*m (median PFS 4.3 versus 2.8 months, (HR 0.57; 95% CI 0.37–0.87; *p* = 0.022) and *BRCA*wt groups (median PFS 5.3 versus 2.8 months, (HR 0.43; 95% CI 0.26–0.71; *p* = 0.0023) [59]. According to the European Society of Gynaecological Oncology (ESGO)–ESMO–European Society of Pathology (ESP) consensus conference recommendations published in the Annals of Oncology in February 2024, patients responding to platinum-based CT after prior PARPi maintenance therapy may be considered for a PARPi maintenance rechallenge, given the duration of a prior PARPi exposure of 18 months in the first line and 12 months in further lines or 12 months and 6 months for patients with a *BRCA*m or *BRCA* wild-type status, respectively [60]. In conclusion, while current evidence supports the feasibility of this approach for certain patients, further research is essential to optimize patient selection, manage and limit resistance and toxicity effects and improve therapeutic outcomes.

## 4. Discussion

PARP inhibitors have revolutionized the treatment landscape for ovarian cancer in a new era of precision medicine. An extensive examination of key clinical trials (SOLO-1, PRIMA, ATHENA-MONO, PAOLA-1) regarding PARPi in first-line maintenance therapy has demonstrated notable improvements in PFS and OS, especially for *BRCA*m and HRD-positive patients [23,24,25,26]. Moreover, the use of PARPi in second-line therapy is also associated with an improvement in terms of PFS, as demonstrated in several trials [55,56,57,58,59,60]. Approximately 20% of OC patients are associated with *BRCA1* and *BRCA2*, while up to 50% present with HRD [9]. For that reason, tumor biomarker testing is fundamental to addressing patients to the most appropriate therapy. Following a consensus among European experts led by Vergote et al., it is recommended to conduct germline and/or tumor *BRCA1/2* testing, as well as HRD tumor testing, at the primary diagnosis stage, preferably before the completion of first-line chemotherapy [43]. The role of MMR testing in OC is still controversial due to the low rate of OC related to Lynch Syndrome, but it could be considered more frequently related to LS in non-serous OC. Despite advancements in more accurate and sensitive biomarker testing, determining the optimal maintenance therapy for advanced ovarian cancer remains a challenge for clinicians both in first- and second-line settings. For instance, prospective trials are essential to establish a universal algorithm for maintenance therapy in advanced OC. Further trials are testing combination strategies with immunotherapy [61,62,63]. However, the role of immune checkpoint inhibitors in ovarian cancer is still controversial [61,62,64].

## 5. Conclusions

In conclusion, the principal challenge in this research field is expanding the target populations, including patients without *BRCA* mutations, across different disease stages. Additionally, there is a need to explore novel combinations of PARP inhibitors with immunotherapies or other targeted agents to enhance treatment outcomes. Finally, overcoming the development of resistance mechanisms, especially in relapsed disease, is also a critical aspect that warrants further exploration. Therefore, further research, such as defining an appropriate treatment course, is needed.

## Figures and Tables

**Table 1 ijms-25-11239-t001:** Description of gene mutations and their functions.

Gene		Function
*BRCA1*	*Breast Cancer 1*	*BRCA1* is involved in DNA repair and the response to DNA damage, playing a crucial role in homologous recombination repair.
*BRCA2*	*Breast Cancer 2*	*BRCA2* is involved in DNA repair and maintaining genomic stability, facilitating homologous recombination repair.
*MRN Complex*	*Meiotic recombination 11-RADiation sensitive protein 50-Nijmegen breakage syndrome 1* *(Mre11-Rad50-Nbs1 complex)*	A protein complex that plays a crucial role in the DNA damage response, especially in repairing double-strand breaks.
*CtlP*	*C-terminal binding protein interacting protein*	A protein that interacts with the *MRN complex* and plays a role in repairing double-strand breaks in DNA.
*MRE11*	*Meiotic recombination 11*	Part of the *MRN complex*, involved in recognizing and repairing double-strand breaks in DNA.
*ATM*	*Ataxia Telangiectasia Mutated*	A kinase that plays a central role in the DNA damage response, activating repair processes and cell cycle checkpoints in response to DNA breaks.
*H2AX*	*H2A Histone Family Member X*	An isoform of the histone *H2A* that is rapidly phosphorylated in response to DNA damage, signaling the presence of double-strand breaks and recruiting repair proteins.
*PALB2*	*Partner and Localizer of BRCA2*	A protein that interacts with *BRCA1* and *BRCA2*, contributing to DNA repair through homologous recombination.
*RPA*	*Replication Protein A*	A protein complex that binds to single-stranded DNA, playing a crucial role in DNA replication and repair.
*RAD52*	*RADiation sensitive protein 52*	A protein involved in DNA repair that facilitates homologous recombination and the pairing of DNA strands.
*MLH1*	*MutL Homolog 1*	A protein in the mismatch repair system, essential for maintaining genomic stability by correcting errors during DNA replication.
*MSH2*	*MutS Homolog 2*	Part of the mismatch repair system, *MSH2* recognizes and binds to DNA mismatches.
*MSH6*	*MutS Homolog 6*	A protein that works with *MSH2* in the mismatch repair system, contributing to the recognition of mismatches.
*PMS2*	*Postmeiotic Segregation Increased 2*	A protein in the mismatch repair system that works alongside *MSH2* and *MSH6*.
*TP53*	*Tumor Protein 53*	*TP53* is a protein that regulates the cell cycle and induces apoptosis in response to DNA damage.

**Table 2 ijms-25-11239-t002:** Description of included studies with main results.

Author (Year)	Study Design	N. Patients	Molecular Biomarker	Test Used	Primary Endpoint	Results
Manchana 2019 [9]	Retrospective	87	*BRCA 1/2*	Next generation sequencing system	Frequency of *BRCA* mutation in epithelial ovarian cancer	Frequency of *BRCA* mutation in high-grade serous carcinoma was 25.7%; none was found in high-grade endometrioid carcinoma
Yang 2011 [19]	Observational study	316	*BRCA 1/2*	Illumina GAIIx platform ABI SOLiD 3 platform	Overall Survival Progression-free survival	*BRCA2* mutation, but not BRCA1 deficiency, is associated with improved survival, chemotherapy response and genome instability compared to *BRCA* wild-type
Bolton 2012 [20]	Meta-analysis	1213	*BRCA 1/2*		Five-year overall mortality	Among patients with invasive epithelial ovarian cancer, having a germline mutation in *BRCA1* or *BRCA2* was associated with improved 5-year overall survival
Candidodos-Reis 2015 [21]	Case–control studies	6556	*BRCA 1/2*	Illumina HiScan sequencer	Effect of germline mutations in *BRCA 1* and *BRCA 2* on mortality	*BRCA1/2* mutations are associated with better short-term survival
Biglia 2016 [22]	Comparative study	A total of 24 with ovarian cancer with a control group of 64 age matched patients with no family history of breast/ovarian cancer	*BRCA 1/2*		Compare clinical–pathological characteristics and outcome between sporadic ovarian cancer and ovarian cancer in patents with hereditary breast and ovarian cancer syndrome	*BRCA*+ patients have a better prognosis than controls in terms of PFS at a median follow-up time of 46 months
González- Martín 2019 [23]	Randomized trial	733	*BRCA*		Progression-free survival in patients who had tumors with homologous recombination deficiency	Those who received niraparib had significantly longer progression-free survival than those who received placebo
Ray Coquard 2019 [24]	Randomized trial	806	*BRCA*		Progression-free survival	The addition of maintenance Olaparib provided a significant progression-free survival benefit
Monk 2022 [25]	Randomized trial	538	*BRCA 1/2*		Progression-free survival	Rucaparib monotherapy is effective as first-line maintenance
Moore 2018 [26]	Randomized trial	391	*BRCA 1/2*		Progression-free survival	Maintenance therapy with Olaparib provided a substantial benefit with regard to progression-free survival
Ramus 2015 [28]	Comparative study	A total of 3374 case patients and 3487 control patients	*BRIP1,* *BARD1,* *PALB2* and *NBN*	Next generation sequencing	Prevalence and EOC risks and evaluated associations between germline variant status and clinical and epidemiological risk factor information	Deleterious germline mutations in *BRIP1* are associated with a moderate increase in EOC risk
Loveday 2011 [29]	Comparative study	A total of 1648 breast ovarian cancer families 1060 controls	*RAD51D*	PCR reactions (Qiagen Multiplex PCR Kit, Quiagen N.V., Hilde, Germany). Amplicons were unidirectionally sequenced using the BigDyeTerminat (Thermo Fisher Scientific, Waltham, MA, USA) or Cycle sequencing kit and an ABI3730 automated sequencer, ABI Perkin Elmer (Thermo Fisher Scientific, Waltham, MA, USA) Mutation Surveyor software (https://softgenetics.com/products/mutation-surveyor/, accessed on 15 October 2024) and by visual inspection. All mutations were confirmed by bidirectional sequencing from a fresh aliquot of the stock DNA	The role of *RAD51D* in cancer susceptibility	*RAD51D* mutation testing may have clinical utility in individuals with ovarian cancer and their families. Moreover, we show that cells deficient in *RAD51D* are sensitive to treatment with a PARP inhibitor
Song 2015 [30]	Case–control study	A total of 3429 patients with invasive EOC 2772 controls 2000 unaffected women who were *BRCA1/BR CA2* negative	*RAD51B,* *RAD51C*, and *RAD51D*	48.48 Fluidigm Access Arrays	Contribution of deleterious mutations in the *RAD51B*, *RAD51C*, and *RAD51D* genes to invasive epithelial ovarian cancer	*RAD51C* and *RAD51D* are moderate ovarian cancer susceptibility genes
Hodgson 2018 [31]	Randomized trial	265	*BRCA1/2*	Next generation sequencing	Long-term outcome of candidate biomarkers of sensitivity to Olaparib in *BRCA* wild-type tumors	Ovarian cancer patients with tumors harboring loss-of-function mutations in HRR genes other than *BRCA1/2* derive treatment benefit from Olaparib similar to patients with *BRCA* mutated
Foglietta 2020 [46]	Prospective observational	A total of 363 probands 50 *BRCA1/2* mutated 28 *BRCA1* 23 *BRCA2*	*BRCA 1/2*	Qubit dsDNA BR Assay Kit	Determine the overall germline *BRCA* variant frequency and spectrum in healthy Italian	Overall frequency of *BRCA* germline variants in the selected high-risk central Italian population is about 13.8%
Pepe 2020 [48]	Retrospective	20	*BRCA 1/2* HRD	SOPHiA DDM HRD Solution, HRD focus Oncomine homologous recombination repair pathway predesigned panel	Technical feasibility, interassay and interlaboratory reproducibility of in-house HRD testing	N-house testing for HRD can be reliably performed with commercially available next- generation sequencing assays
Kfoury 2023 [49]	Prospective study	25	*TP53* *BRCA* HRD	Next generation sequency SNP array	Evaluate the feasibility and usefulness of HRD testing on cfDNA from ascites	Copy number analysis on ascitic cfDNA is feasible and can be used to detect the same HRD scar as tumor testing
Banerjee 2021 [50]	Randomized trial	391	*BRCA*		Progression-free survival	The longest follow- up for any randomized controlled trial of a PARP inhibitor in this setting, the benefit derived from 2 years’ maintenance therapy with Olaparib was sustained beyond the end of treatment, extending median progression-free survival past 4.5 years
DiSilvestro 2023 [51]	Randomized trial	391	*BRCA*		Overall survival after 7-year follow-up	Clinically improvement in OS with maintenance. Olaparib in patients with newly diagnosed advanced ovarian cancer and a *BRCA* mutation
RayCoquard 2023 [52]	Randomized trial	228	*BRCA* HRD		Prespecified final overall survival analysis, including analyses by HRD status	Olaparib plus bevacizumab provided clinically meaningful OS improvement for first-line patients with HRD-positive ovarian cancer

*BRCA*: breast cancer susceptibility gene; PFS: progression-free survival; EOC: epithelial ovarian cancer; HRD: homologous recombination repair deficiency; *TP53*: tumor protein 53.

## Data Availability

The data presented in this study are available within the article.

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
