# Peer review of "The Role of Tumor Biomarkers in Tailoring the Approach to Advanced Ovarian Cancer"

_ijms, 2024, doi:10.3390/ijms252011239_

Round 1
Reviewer 1 Report
Comments and Suggestions for Authors
First and foremost, the article needs to be edited to reduce similarity. Looks like some sentences have been copy and pasted from existing literature.
Since this is a Review article, there should not be sections such as Results!
Section 2.1 through 2.5 can make up section 2 but the information provided in section 2.6 pertains to clinical trials - this needs to be a new section – section 3
Further, this section 2.6 is one long paragraph – this is quite monotonous. As mentioned above, make it a new section and then arrange this into few small sub-sections depending on the information discussed. Perhaps the individual sub-sections can discuss clinical trials for different mutations.
2.7 can also be a sub-section of new section 3.
Again, since this is a Review article, there is no need for ‘Materials and Methods’.
One major concern is that there is just one Table in the entire manuscript. Authors can think of one cartoon diagram to summarize the mutations and the associated clinical trials.
Author Response
Response to Reviewer 1:
Comment 1R1: First and foremost, the article needs to be edited to reduce similarity. Looks like some sentences have been copy and pasted from existing literature.
Response 1R1: We thank the reviewer for this observation. We carefully revised the manuscript to reduce the similarities with the existing literature. We rephrased sentences and ensured that all paraphrased sections were appropriately cited. Additionally, we conducted a thorough review to guarantee that the text reflects original thought while still acknowledging relevant sources.
Comment 2R1: Since this is a Review article, there should not be sections such as Results!
Response 2R1: Thank you for your valuable feedback regarding the structure of the manuscript. In response to your comment, we have modified the article by removing the "Results" heading. We have also added new sections as suggested with standalone titles to enhance the narrative flow of the review.
Comment 3R1: Section 2.1 through 2.5 can make up section 2 but the information provided in section 2.6 pertains to clinical trials - this needs to be a new section – section 3.
Response 3R1: Thank you for your valuable opinion on the organization of the manuscript. We have consolidated Section 2 under the title “Exploring DNA Repair Pathways: The Impact of BRCA1/2 and Other Mutations on Cancer Susceptibility and Treatment,” while maintaining the subsections based on the various mutations evaluated. Additionally, we have established a new Section 3, which contains subsections focused on first-line and second-line therapy, specifically addressing the pivotal results of clinical trials.
Comment 4R1: Further, this section 2.6 is one long paragraph – this is quite monotonous. As mentioned above, make it a new section and then arrange this into few small sub-sections depending on the information discussed. Perhaps the individual sub-sections can discuss clinical trials for different mutations.
Response 4R1: Thank you for your valuable feedback on section 2.6. In response, we have merged sections 2.6 and 2.7 into a new section 3 titled “Clinical Outcomes in Homologous Recombination Deficiency: Key Insights from Clinical Trials in First-Line and Second-Line Therapies.” This section is now divided into several smaller sub-sections, each focusing on specific clinical trials related to different PARP inhibitors, biomarker backgrounds, and first- or second-line settings. This restructuring ensures a more organized and engaging presentation of the data. We appreciate your suggestion, which has significantly improved the clarity and flow of the manuscript.
Comment 5R1: 2.7 can also be a sub-section of new section 3.
Response 5R1: Thank you for this comment. We have established a new Section 3 under the title “Clinical Outcomes in Homologous Recombination Deficiency: Key Insights from Clinical Trials in First-Line and Second-Line Therapies”, which contains several subsections focused on first-line and second-line therapy, specifically addressing the pivotal results of clinical trials.
Comment 6R1: Again, since this is a Review article, there is no need for ‘Materials and Methods’.
Response 6R1: Thank you for your valuable feedback regarding the structure of the manuscript. In response to your feedback, we have removed this section to ensure the manuscript conforms to the standard format for review articles.
Comment 7R1: One major concern is that there is just one Table in the entire manuscript. Authors can think of one cartoon diagram to summarize the mutations and the associated clinical trials.
Response 7R1: Thank you for your insightful suggestion. In response, we have decided to add a new table that summarizes the various mutations discussed in the text along with their functions. Additionally, the existing table in the manuscript already provides information on the type of biomarker analyzed for each study.
Reviewer 2 Report
Comments and Suggestions for Authors
Very well written paper.
I would recommend to add a table with the mentioned proteins/mutations next BRCA, e.g. RAD51, PALB2 and their function to make it easier to assess the individual influences of these components. You also use quite a few abbreviations such as MLH1, MSH2 not including their full name, might be done with such a table to ease the reading of the text.
Furthermore, as the current table 1 is spreading over a few pages, it would be good to include the titles of the individual columns on the next page. Would make it smoother to read.
Altogether very well written and a good summarization on how biomarker scanning is important for therapeutic approaches.
Author Response
Response to Reviewer 2:
Comment 1R2: Very well written paper. I would recommend to add a table with the mentioned proteins/mutations next BRCA, e.g. RAD51, PALB2 and their function to make it easier to assess the individual influences of these components. You also use quite a few abbreviations such as MLH1, MSH2 not including their full name, might be done with such a table to ease the reading of the text.
Response 1R2: We are honored that you appreciated our work. Thank you for your valuable suggestion. In response, we have added a table that outlines the proteins and mutations mentioned, along with a brief description of their functions. This will facilitate readers in evaluating the individual influences of these components. Additionally, the table will include the full names of the abbreviations used, enhancing clarity and improving the overall readability of the manuscript.
Comment 2R2: Furthermore, as the current table 1 is spreading over a few pages, it would be good to include the titles of the individual columns on the next page. Would make it smoother to read.
Response 2R2: Thank you for your insightful suggestion regarding Table 1. We agree that having the column titles repeated on subsequent pages will improve readability. In response, we included the titles of the individual columns on the following pages where the table continues.
Comment 3R2: Altogether very well written and a good summarization on how biomarker scanning is important for therapeutic approaches.
Response 3R2: Thank you for your positive feedback! We appreciate your recognition of this critical aspect of our work.
Reviewer 3 Report
Comments and Suggestions for Authors
The main question addressed by the research is related to the recent literature on the correlation between tumor biomarkers and chemotherapy in nonmucinous ovarian cancer, providing suggestions for personalised treatment approaches.
The topic is relevant to the field and it addresses a specific gap in the field which is related to the fact that appropriate treatment of ovarian cancer does not only related to the stage and other “classic” parameters.
Compared with other published material, it adds the particular focus on the correlation between tumor biomarkers and chemotherapy in nonmucinous ovarian cancer.
Regarding the methodology, some more data are necessary about related syndromes, as Peutz-Jeghers syndrome, associated with pathogenic variants in the Serine/Threonine Kinase 11 (STK11) gene.
The conclusions are consistent with the evidence and arguments presented and they address the main question posed because they summarise the relevant literature.
The references are appropriate.
Table 1 is useful because it summarises the included studies with the main results.
Author Response
Response to Reviewer 3:
Comment 1R3: The topic is relevant to the field and it addresses a specific gap in the field which is related to the fact that appropriate treatment of ovarian cancer does not only related to the stage and other “classic” parameters.
Response 1R3: We are pleased to hear that you find the topic relevant and that it addresses a specific gap in the field. We agree that the treatment of ovarian cancer extends beyond traditional parameters, and our research aims to highlight the importance of tumor biomarkers in guiding personalized treatment approaches.
Comment 2R3: Compared with other published material, it adds the particular focus on the correlation between tumor biomarkers and chemotherapy in nonmucinous ovarian cancer.
Response 2R3: Thank you for acknowledging the unique focus of our study on the correlation between tumor biomarkers and chemotherapy in non-mucinous ovarian cancer. We believe that this perspective is crucial for advancing personalized medicine in this area.
Comment 3R3: Regarding the methodology, some more data are necessary about related syndromes, as Peutz-Jeghers syndrome, associated with pathogenic variants in the Serine/Threonine Kinase 11 (STK11) gene.
Response 3R3: We appreciate your suggestion to include additional data on syndromes such as Peutz-Jeghers syndrome (PJS). However, ovarian cancers associated with PJS are typically sex-cord stromal tumors, which are nonepithelial in origin and generally benign and not suitable for maintenance therapy. Given that the focus of our review is on the key genetic mutations in epithelial non-mucinous ovarian cancer and their potential role in therapy decisions—particularly in the context of maintenance treatments with PARP inhibitors or immune checkpoint inhibitors—we believe that including a section on PJS would be outside the scope of this review.
Comment 4R3: The conclusions are consistent with the evidence and arguments presented and they address the main question posed because they summarise the relevant literature.
Response 4R3: Thank you for your positive feedback on our conclusions.
Comment 5R3: The references are appropriate.
Response 5R3: We are glad you found our references appropriate. We aimed to include a comprehensive selection of literature that supports our findings and recommendations.
Comment 6R3: Table 1 is useful because it summarises the included studies with the main results.
Response 6R3: Thank you for your feedback on Table 1.
Round 2
Reviewer 1 Report
Comments and Suggestions for Authors
The manuscript has been ideally revised.